# Building a MultimodalClassifier of Email Behavior: Towards a Social Network Understanding of Organizational Communication

Harsh Shah [1,†], Kokil Jaidka [2,*,†], Lyle Ungar [3], Jesse Fagan [4] and Travis Grosser [5]

1    Department of Electrical & Electronics Engineering, Birla Institute of Technology Pilani, Rajasthan 333031, India
2    Department of Communications and New Media, National University of Singapore,
     Singapore 119077, Singapore
3    Department of Computer and Information Sciences, University of Pennsylvania, Philadelphia, PA 19104, USA
4    The Business School, University of Exeter, Exeter EX4 4PY, UK
5    The School of Business, University of Connecticut, Storrs, CT 06269, USA
*    Correspondence: jaidka@nus.edu.sg
†    These authors contributed equally to this work.

**Abstract:** Within organizational settings, communication dynamics are influenced by various factors, such as email content, historical interactions, and interpersonal relationships. We introduce the Email MultiModal Architecture (EMMA) to model these dynamics and predict future communication behavior. EMMA uses data related to an email sender's social network, performance metrics, and peer endorsements to predict the probability of receiving an email response. Our primary analysis is based on a dataset of 0.6 million corporate emails from 4320 employees between 2012 and 2014. By integrating features that capture a sender's organizational influence and likability within a multimodal structure, EMMA offers improved performance over models that rely solely on linguistic attributes. Our findings indicate that EMMA enhances email reply prediction accuracy by up to 12.5% compared to leading text-centric models. EMMA also demonstrates high accuracy on other email datasets, reinforcing its utility and generalizability in diverse contexts. Our findings recommend the need for multimodal approaches to better model communication patterns within organizations and teams and to better understand how relationships and histories shape communication trajectories.

**Keywords:** email; organization; social network analysis; text classification; computational linguistics; transformers

## 1. Introduction

In today's digital era, emails dominate professional communication. Their importance grew even more during the 2020–2021 remote work shift. However, despite their widespread use, there is a limited computational understanding of professional email behavior, leading to costly inefficiencies. Studies show US workers spend nearly half their time on emails, with 86% facing communication challenges [1]. A study of enterprise email logs reported that 12–16% of emails are deferred daily [2]. Remote teams lose about 7.5 h weekly due to these issues. Such lapses in communication disrupt the workflow and have tangible financial repercussions, with businesses incurring costs of over 12,000 USD annually per employee due to poor communication.

A formal study of professional email communication could propel the training and deployment of deep learning models that lead to productive email behavior, such as predicting when an email will get a reply and, as a corollary, detecting and responding to urgent emails promptly. In this context, it is important to consider the role of social networks in online organizational behavior. Professional networks are indicators of influence and power dynamics and are thereby important signals of interpersonal behavior. Studies of workplace teams have emphasized the role of networked peers in participation, knowledge creation, and information dissemination [3,4].

However, we find that there is a discernible gap in incorporating social network features when understanding and modeling email communication behavior. Few studies have leveraged social network analyses to model organizational communication behavior, and one previous study examined the patterns of resource exchange with a similar dataset [5]. Some of these studies highlighted the role of brokers in workplace communication. Brokers are individuals bridging structural gaps between social groups [6]. These brokers, due to their diverse information exposure, are believed to have a 'vision advantage' and play a pivotal role in small-world network evolution [5–10]. However, there is ambiguity about whether brokers hold any influence in email communication. On the one hand, brokers have access to diverse information, which can translate into significant influence compared to central figures in cohesive groups. Conversely, brokers might wield less influence than central figures within cohesive groups [5].

This study addresses the research gap in incorporating social network features in a traditional text classification problem, with a multi-modal representation of organization behavior to contextualize email reply behavior. In doing so, it will answer several questions focusing on the role of individual and interpersonal effects in professional email communication between a sender and a receiver.

*Research Objectives*

This study aims to develop a model that can predict the probability of receiving a reply to an email. The model will be trained on a dataset of email conversations. We consider whether or not an email received a reply as our dependent variable based on subsequent emails in the dataset. The features used to train the deep learning models will be the text of the email, the sender's writing style, the receiver's writing style, the relationship between the sender and the receiver, and other features, both implicit and explicit, which have been explained in detail in further sections. We ask:

- RQ1 How do the sender's (a) professional influence and (b) personal influence in the organizational network affect their likelihood of receiving a reply?
- RQ2: How well does a pretrained transformer finetuned with multimodal features predict email replies?
- RQ3: How does its performance vary with (a) different feature sets in the model and (b) different training dataset sizes?
- RQ4: How does its performance vary for different datasets?

To address these research gaps, we developed and tested a multimodal transformer architecture for predicting email responsiveness in an organization. Recent work in communication computational methods has evinced interest in using deep learning models for text classification [11], while recent studies focusing on reply behavior have expanded to consider the context of online dating [12]. Some studies have explored social network analyses to model reply behavior in the workplace [9,10]. On the other hand, fewer papers have explored building multimodal classifiers for text classification [13]. Building on prior work, in this study, we propose EMMA, an Email MultiModal Architecture, which trains deep learning models to predict whether or not an email will receive a reply in terms of the linguistic, organizational, and interpersonal context it embeds.

Our work has two main contributions. First, we illustrate how style and language is secondary to social influence for text classification, even with transformer-based models, which have so far focused on only text or multimedia features. Reinterpreting the traditional understanding of 'multimodal' data, we enrich our inputs with information about the social context of the email and demonstrate remarkable improvements in email reply prediction. Second, we show that our approach is practically feasible for other organizational contexts as we validate the importance, robustness, and generalizability of the multimodal feature representation on two other standard email datasets—the Enron dataset and the Avocado dataset. Across all three datasets, we show that EMMA offers a significant predictive advantage over simpler transformer architectures for the problem of email reply prediction.

Through our experiments we illustrate the importance of social influence and linguistic accommodation for predicting enterprise email replies using transformer-based models.

Our research also provides nuanced theoretical perspectives on individual roles within organizational networks and the distinct markers that define interpersonal relationships in professional email communication. The psychological ramifications of email usage, a subject of keen interest to many scholars, are further illuminated by our findings, suggesting avenues for integrating these insights into organizational email practices.

## 2. Related Work

In recent years, email data have emerged as a pivotal resource for understanding organizational dynamics, offering insights into information flow, stylistic intricacies, and broader implications for communication and organizational culture [14–20].

A significant portion of prior research has been dedicated to predicting email responses. Classical machine learning models have been the mainstay, often augmented with features capturing social interactions [14,16,21]. For instance, a study [22] proposed a model to gauge the likelihood and time frame within which a recipient might respond to an email, emphasizing the relevance of features such as the number of attachments and the length of the email body. Another innovative approach by [23] involved a content-based recommendation system, which gauged the similarity between incoming emails and users' existing inboxes, enhancing prediction accuracy.

However, a common thread across these studies is a certain level of oversight. Many have overlooked the nuanced interplay of interpersonal dynamics and linguistic accommodation inherent in one-on-one communication. Moreover, the potential of deep learning in predicting email responses still needs to be explored. Existing models have focused on summarizing the knowledge in emails [24,25]. For instance, the study by Jörgensen [24] applied deep learning to extract knowledge from email networks [24]. More recently, the emergence of billion-feature representations offers the potential for the richer modeling of email data; however, they are still unsuitable to the sender's professional and personal influence. To bridge this gap, we introduce a multimodal architecture. This innovative approach, built upon a modified RoBERTA framework, seamlessly integrates numerical and categorical features, offering a more holistic representation of emails.

Shifting the focus to linguistic styles, previous studies have employed professional communication datasets to discern patterns such as email formality [26], confidentiality [27], and topical relevance [28,29]. Extracting tasks and intents from email has also been a focal point of much NLP research [22,30,31]. Deep learning, while a powerful tool, has been primarily applied to problems such as email sequencing [14,16,32] and predicting the sender's gender [33], while more recently, the inclusion of natural language generation capabilities in email servers and software has spurred studies of email reply suggestions [34,35].

Historically, the Enron email dataset has been the cornerstone for such research. However, recent endeavors have combined the Enron corpus with the Avocado corpus [36], leveraging advanced language models and contextual word embeddings. These combined resources have been instrumental in predicting the nature of emails [37] and understanding their intent and significance [22]. Fewer studies in NLP have worked with the Luxury Standard dataset, which, apart from email histories, encompasses features that illuminate the personal influence of employees, enabling a comprehensive evaluation of various influence mechanisms within professional settings [38].

### 2.1. Email Behavior

The literature on email behavior spans various topics, from personality-driven interactions to the psychological ramifications of email usage. Many of these studies specifically deal with how email usage affects how individuals interact and are affected. One prominent study area focuses on the relationship between personality differences and email activity. For instance, Ref. [39] delved into how personality differences can predict action–goal relationships in work-email activity, emphasizing the role of individual differences

in underpinning one's choice of strategy in multi-goal work environments [39]. Similarly, ref. [40] explored the connection between personality and the phenomenon of email overload and its subsequent impact on burnout and work engagement. Another area of research centers on decision-making styles and their influence on email use. Ref. [41] found that avoidant decisional styles, such as procrastination and buck-passing, could predict higher levels of email use in the workplace. This is complemented by the work of [42], who described various strategies individuals employ when dealing with email interruptions, highlighting the situational parameters that influence these strategies. While these findings suggest the importance of individual differences in predicting workplace behavior, such as the likelihood of replying to professional emails, these ideas have yet to be factored into predictive models of email reply behavior.

Our work is most related to the emerging third area of email-related research, which delves into the nuances of email communication features and their impact on team dynamics. Prior work on email behavior has focused on both marketing and professional emails. Studies that predict whether recipients will open marketing emails have focused on measuring the sentiment in the subject line [43] as well as other linguistic features and explored the need to adapt to new domains [44]. On the other hand, in professional communication, email responses have been predicted based on linguistic signals [45,46]. However, only some studies have done so for email reply prediction; furthermore, these models may miss other non-linguistic signals, such as who is sending the email and their impression of the receiver. Ref. [47] provided insights into how the use of email features such as Cc, Bcc, forward, and rewrite can significantly influence team dynamics. This is juxtaposed with ref. [46], which investigated the role of workload and civility in email communication, finding that participants were more likely to respond with incivility to uncivil stimuli, especially under high workloads [46]. The findings suggest the importance of considering directed email separately from all other kinds of email and the tendency to match linguistic styles with the receiver—two ideas that motivated our research design and feature exploration.

### 2.2. The Role of the Social Network

Social networks serve as invaluable reservoirs of knowledge, support, influence, and validation, embodying a form of social capital. Social networks matter in the study of coordination behavior and group efficacy [48]. In the realm of professional communication, the significance of networks in fostering effective team communication has been well-documented [17,49]. For instance, Ref. [49] posited that while top-down communication tends to dictate behavior, bottom-up communication can foster group-level coordination. Delving deeper into the structural intricacies of professional communication, studies have probed the correlation between perceived measures of influence and one's positional influence within the communication network [6]. Such positional influence not only accrues social capital but also tangibly impacts real-world outcomes, such as job referrals [50].

The "modern weak tie theory" [15] is central to our exploration. This theory underscores the importance of strong ties, characterized by frequent communication, in fostering intra-group cohesion in dynamic work settings. Building upon the foundational work on weak ties [51], we emphasize the pivotal role of brokers, or individuals bridging communication gaps, in maintaining and transcending team boundaries. In this regard, we examine how the different measures of an employee's professional and personal influence contribute to predicting future behavior, such as their likelihood of receiving a response to their email. Our focus narrows to understanding how various measures of an employee's professional and personal influence can predict future behaviors, such as their propensity to receive an email response. Within our datasets, an individual's *professional influence* is gauged by their centrality in organizational communication [6] and their potential for promotion, as reflected in performance evaluations [52]. Conversely, their *personal influence* encapsulates their likability among peers, discerned both through direct interpersonal ratings [53] and inferred through mirrored behaviors in email exchanges [54]. We contend that integrating

these diverse facets of social capital can offer a clearer picture of influential figures within email communication.

In summation, while the text is a vital component, it alone falls short in capturing the intricate interpersonal dynamics pivotal to understanding the email reply prediction challenge. This realization paves the way for our proposed Email MultiModal Architecture (EMMA), a model that synergizes textual email features with the sender's style, social influence, and the linguistic rapport between the sender and recipient, thereby enhancing model performance.

## 3. Method

We focus on the problem of predicting whether or not an email will receive a reply. The task is binary classification with fine-tuned transformers, where a 0 implies not receiving a reply and 1 means receiving a reply to the mail $e_t$. We trained deep learning models on four types of features: stylistic features, professional influence features, personal influence features, and linguistic accommodation features. Methodologically, we offer a novel approach that interprets social network features (professional influence features) as "multimodal" in finetuning BERT models, an approach that addresses the lack of exploration around network-enriched transformer methods in text classification or for the multimodal representation of text. We find that our architecture offers predictive advantages across three different organizational datasets. Next, we generalize the findings to two other well-known email datasets, where once again, we demonstrate how multimodal feature representation offers a significant advantage in transformer-based models. Our experiments illustrate the importance of social influence and linguistic accommodation for predicting enterprise email replies using transformer-based models.

### 3.1. Framework

The EMMA end-to-end finetuning pipeline is shown in Figure 1. We represent every interaction between a sender and a recipient as a set of features input into an end-to-end multimodal RoBERTa-based transformer. The *Text Encoder Layer* encodes the email body as finetuned RoBERTa embeddings. RoBERTa can handle the email body and generate contextual embeddings for the mail body. To handle the extra non-textual features, some architectural changes need to be made to the model rather than simply adding a few layers after RoBERTa and finetuning it. Therefore, the *Gating Layer* combines the stylistic, social network, influence, and linguistic accommodation features with the 768-dimension embedding output by the RoBERTa model. Finally, this combined embedding, which contains the overall email representation, is fed to a linear layer that predicts the likelihood of receiving a reply. Further details on the framework are discussed below.

#### 3.1.1. Text Encoder Layer

We applied the pre-trained BERT model and RoBERTa models (base uncased) to represent email features as a 768-embedding feature set. The hyperparameters are reported in the Experimental Setup section.

#### 3.1.2. Gating Layer

We use the gating mechanism that has been proposed and implemented for multimodal BERT architectures [55]. The combined multimodal features $m$ are the weighted sum of textual features $x$ with the transformed non-textual features $h$ as below:

$$\mathbf{m} = \mathbf{x} + \alpha\mathbf{h} \tag{1}$$

The scaling factor $\alpha$ ensures that the effect of the transformed non-textual features $h$ does not overpower the textual features $x$. $\alpha$ is calculated as a ratio of the $L_2$ norm of $x$ and $h$ as:

$$\alpha = \min\left(\frac{\|\mathbf{x}\|_2}{\|\mathbf{h}\|_2} * \beta, 1\right) \tag{2}$$

where $\beta$ is a hyperparameter. Both the equations above refer to the transformed feature matrix $h$, which is the output of a multi-layer perceptron that implements a gating mechanism on $n$ (numerical features) and $c$ (categorical features).

We calculate $h$ using the following equation:

$$\mathbf{h} = \mathbf{g_c} \odot (\mathbf{W}_c\mathbf{c}) + \mathbf{g_n} \odot (\mathbf{W}_n\mathbf{n}) + b_h \tag{3}$$

Here, $W_c c$ and $W_n n$ are the categorical and numerical feature matrices, respectively. The gating vector $g_i$ is computed using the following method:

$$\mathbf{g_i} = R(\mathbf{W_{gi}}[\mathbf{i}|\mathbf{x}] + b_i) \tag{4}$$

where $\mathbf{W_{gi}}$ is the full feature matrix with textual and non-textual features, $i$ denotes the numerical and categorical features, and $R$ is a non-linear activation.

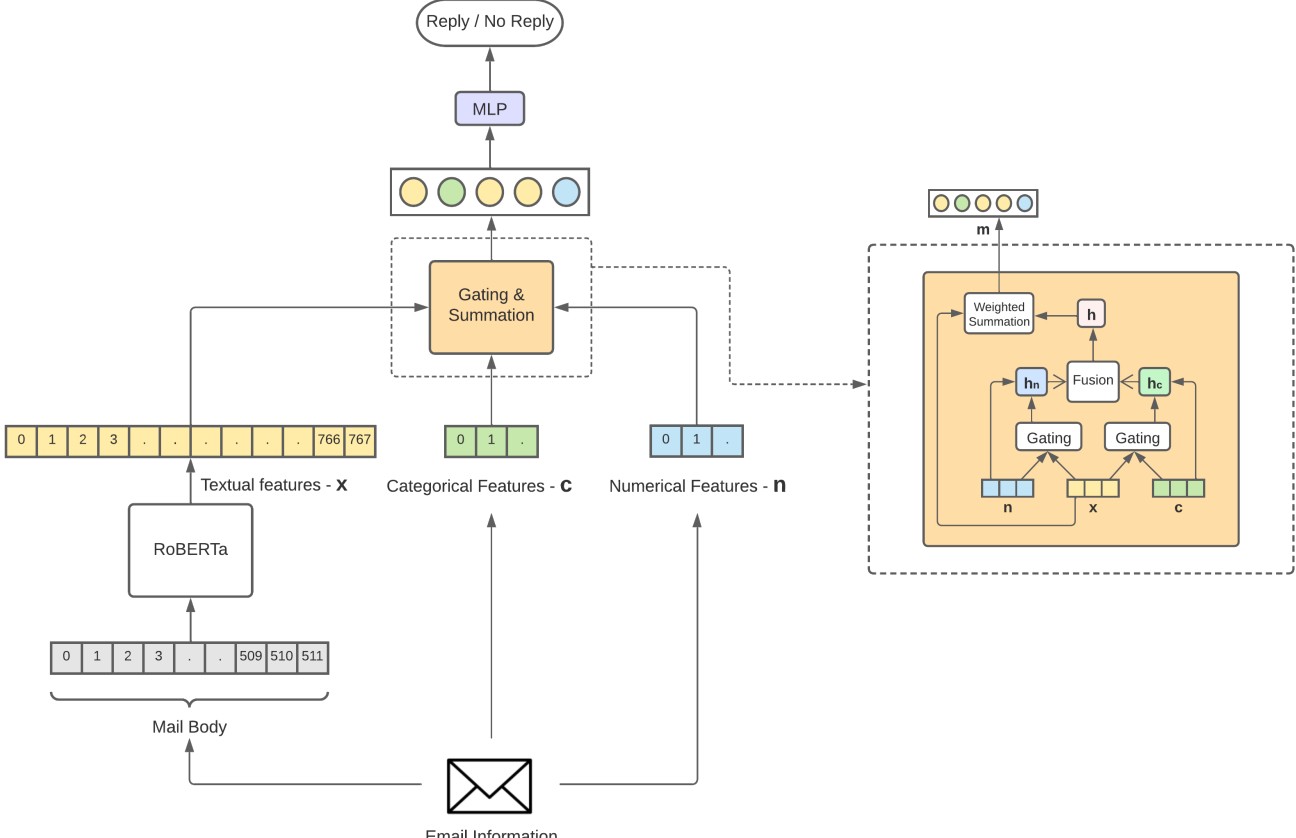

**Figure 1.** The proposed Email MultiModal Architecture.

### 3.2. Experimental Setup

We performed the following experiments to test and validate the EMMA framework:

- **Contribution analysis:** To address RQ1a and RQ1b, we explored the association of individual measures with reply prediction by fitting quasibinomial generalized linear models to our dataset after standardization and feature selection and visualizing the feature importance in terms of the magnitude of the coefficients in the final fitted model.

- **Cross-validation:** To address RQ2, we used the Luxury Standard dataset to train models on the stylistic, professional, personal, and accommodation features just discussed for the email reply prediction task using a cross-validation setup with a 20% held-out test set. We benchmarked the performance of a multimodal transformer against several non-transformer and transformer baselines.

- **Ablation analysis:** To address RQ3 and establish the robustness of EMMA, we conducted an ablation analysis with different input feature sets (RQ3a) and dataset sizes (RQ3b).
- **External validity:** To address RQ4, we also evaluated EMMA on the Enron and Avocado datasets for the same problem, thus ensuring that EMMA generalizes to new data and social contexts.

### 3.3. Baselines

We evaluated the performance of EMMA against a variety of deep learning architectures. All these models were trained on a Tesla V100-SXM3 with a VRAM of 32GB.

### 3.3.1. Non-Transformer Models

We experimented with CNN [56], LSTM [57], BiLSTM [58] and LSTM with attention [59]. The inputs for all these models were the 100-dimension GloVe [60] word vectors as the embedding layers, which were then input into their respective architectures. The models were trained by the ADAM optimizer [61]. These are discussed below.

- **CNNs:** Convolutional neural networks (CNNs) are a class of neural networks that are used primarily for image recognition and classification and were first developed for optical character recognition (OCR)-related tasks [62]. Each layer in a CNN has filters that slide on the data from the input/previous layer. In the case of text, CNN's sliding window captures patterns in the sequence of words, which become more complex with more convolution layers. In our experiments, in the CNN framework [56], we applied convolutional filters followed by max-over-time pooling to the word vectors for a single email.
- **RNNs:** Recurrent neural networks (RNN) are a class of neural networks that is primarily used for sequential data, such as text, audio, and time series data. While neurons in a vanilla DNN can look at input data or data from the previous layer, the neurons in an RNN can go one step further. They can look at both neurons from the last layer and the neuron output from the previous timestep. This small change in RNNs allows them to retain information from the past while calculating the output at a given timestep. To update weights in RNNs, backpropagation through time, or BPTT, is used. The derivation of the weight update rule in RNNs shows that for an input sequence length $= N$, the derivative term contains a weight raised to the power $N$. This means that if the weight term is either very small or very large, the gradient will tend to zero or explode, respectively. Furthermore, the model performance deteriorates with increasing input length.
- **LSTMs:** Architectures such as long short-term memory (LSTM) addressed and corrected the shortcomings in RNNs related to the vanishing gradient at very small or very large weights [57]. Apart from the primary recurrence relation present in RNNs, LSTMs also have multiple gating mechanisms, allowing them to choose the amount of information from the past that is carried forward. An LSTM comprises a forget gate, an input gate, and an output gate. The forget gate controls the amount of information that is forgotten, the update gate controls the amount of information carried forward, and the output gate controls the amount of information passed on to the next layer.
- **LSTM + Attention:** Attention mechanisms address the problems with RNNs that occur with increasing input length. The multi-head self-attention mechanism used in the transformer architecture [63] involves a self-attention component that looks at each word in the input sequence and computes how important other words in the input are when computing the representation for that particular word. A dot product computes the similarity between the current word and prior words in the input sequence, while a key matrix scales down the dot product so that the gradient does not become unstable for large inputs. A softmax function converts the similarity to a probability distribution, adding up to 1. The final multiplication with the value matrix results in those words retaining their embedding whose dot product score is

high, which results in meaningful and contextual embeddings. Transformers have multiple such heads, the outputs of which are concatenated and transformed into a compatible dimension. Another feature of transformer models is that they use multiple encoders and decoders stacked on top of each other as their encoder and decoder blocks.

- **Training hyperparameters**: We trained the models on 15 epochs with a batch size of 1024 and early stopping based on validation loss. The training data were split into training and validation sets during each epoch with a 0.8:0.2 split. This was done to monitor the validation loss. Early stopping was enabled, which stopped the training if the validation loss did not improve for three consecutive epochs. All the DL-based baseline models were trained with following configuration:

  - Batch size = 1024;
  - Vocabulary size = 2000;
  - Input length = 200;
  - Word embedding dimension = 50;
  - Number of epochs = 15;
  - Train:test split = 0.8:0.2.

Apart from this, during each epoch, the training data were split into training and validation sets with a 0.8:0.2 split. This was carried out to monitor the validation loss. Early stopping was enabled, which stopped the training if the validation loss did not improve for 3 consecutive epochs.

### 3.3.2. Transformer Models

Among transformers, we experimented with a pre-trained BERT model [64] and the Robustly optimized BERT model (RoBERTa) [65].

- **BERT:** Bidirectional Encoder Representations from Transformers (BERT) was originally published by Google AI Language in 2018, where they used the transformer architecture for the task of language modeling [64]. One of the key features of BERT is that it stacks only encoder blocks on top of each other. BERT is deeply bidirectional, i.e., the model learns from left-to-right and right-to-left while going over input sentences. Also, the self-attention combined with multi-head attention in each transformer encoder block helps to generate more contextual embeddings by taking into account other words in the sentence when encoding a particular word. BERT models have multiple attention heads to focus on different aspects of the input text simultaneously. Their deep bidirectional nature allows them to infer context from before and after a word within an input text. Furthermore, the combined training task of masked language modeling and next-sentence prediction is also one of the reasons why the BERT language model has a good understanding of the structure and semantics of the English language. Due to these reasons, a single dense layer on top of the output of the BERT language model performs exceptionally well on many NLP tasks.

- **RoBERTa:** Robustly Optimized BERT Pretraining Approach (RoBERTa) is an alternate approach to training BERT that not only improves its performance but also results in easier training [65]. BERT masks tokens randomly during the preprocessing stage, resulting in a static mask. RoBERTa applies a dynamic masking scheme, meaning masking takes place before a sentence is selected into a minibatch to be fed into the model. This approach avoids the pitfall of using the same masked sequence every epoch.

- **Training hyperparameters:** For BERT and RoBERTa, the following training arguments were used:

  - Batch size = 150;
  - Input length = 200;
  - Number of epochs = 10;
  - Initial learning rate = $5e^{-5}$;

- Warmup ratio = 0.07;
- Weight decay = 0.5.

The loss function used to train all these models is binary cross-entropy, which quantifies the difference between the predicted probabilities and actual binary outcomes (0 or 1). It penalizes predictions that are confident and wrong more heavily than predictions that are less confident.

$$L(y, p) = -(y \log(p) + (1 - y) \log(1 - p)) = \text{Loss Function}$$
$$y = \text{Actual Class}$$
$$p = \text{Predicted Class}$$

For an optimizer, the ADAM optimizer [61] was used for all baseline DL models, and ADAM with weight decay was used for transformer baseline models.

## 4. Data Collection

**Luxury Standard**. Our primary analysis is based on a new dataset of 0.6 million timestamped corporate emails exchanged by 4320 employees in 2012–2014, explored in recent works on organization behavior [38,52,66]. A subset comprising around 23,000 emails includes interpersonal ratings by employees about their colleagues on qualities such as friendship, mentorship, problem-solving, collaboration frequency, and avoidability.

**Enron**. The Enron email corpus comprises emails sent over 3.5 years (1998 to 2002), which were made public during the US government's legal investigation of Enron. It was first released as a corpus for computational linguistic analysis in 2004 [67].

**Avocado**. The Avocado dataset comprises a dataset of emails from an anonymous defunct information technology company referred to as Avocado, which the Linguistic Data Consortium released in 2015 [36].

Dataset statistics are reported in Table 1. We constructed email threads for each dataset that grouped emails in sender–receiver pairs. Following the approach by [68], we considered the dyadic email exchange between two individuals to constitute an independent 'thread' of conversation. We labeled each email in a thread according to whether or not it received a reply (i.e., whether the recipient sent a subsequent email in the same thread). From our primary dataset, we selected all the emails sent during one year, while for the secondary datasets, we randomly sampled a 1:1 dataset out of the total emails sent in two years. We restricted our dataset to threads comprising at least ten emails or more to avoid the flooring effects of one-off email exchanges.

**Table 1.** Datasets Description.

|  | Luxury Standard | Enron | Avocado |
| --- | --- | --- | --- |
| Emails exchanged | 665,120 | 96,409 | 199,763 |
| Replies received | 555,645 | 86,332 | 187,748 |
| Employees | 4320 | 8403 | 1041 |
| Time period in days | 443 | 730 | 730 |
| E-mails per employee | 154 | 21 | 191 |
| E-mails per day | 1501 | 132 | 273 |

### 4.1. Feature Extraction

In this study, we combined the stylistic, social network, influence, and linguistic accommodation features with the 768-dimension vector output by the RoBERTa model and finetuned this combination in the end-to-end EMMA architecture. The following paragraphs describe this process in detail.

4.1.1. Stylistic Features (93 Features)

Psycholinguistic features in writing have been used in previous work to identify synchrony and linguistic style matching, including in professional correspondence [69]. We used the Linguistic Inquiry and Word Count tool [70] and focused mainly on four categories of features that are conceptually relevant to modeling email correspondence (and their likelihood of receiving responses):

- **Cognitive processes**: Measures of cognitive activity through evidence of words that denote insight (*think*, *know*), causation (*because*, *effect*), discrepancy (*should*, *would*), tentative (*maybe*, *perhaps*), certainty (*always*, *never*), differentiation (*hasn't*, *but*, *else*) and perception (*look*, *heard*, *feeling*).
- **Emotional features**: Measures of affect in writing, comprising measures of positive emotion (*love, nice, sweet*) and negative emotion (*terrible, worried, sad*).
- **Drive features**: Measures denoting what motivates people, offering insights into perspectives on achievement (*win, success, better*), affiliation (*ally, friend, social*), a need for domination (*superior, better*), and finally, the reward-(*take, prize, benefit*) or risk-orientation (*danger, doubt*) of the sender.
- **Informal language features**: Measures of the casualness of the email body, denoted by the use of swear words (*damn, shit, hell*), netspeak (*lol, thx, lmao*), nonfluencies (*err, hmm, umm*), and fillers (*well, you know, I mean*).

4.1.2. Professional Influence (2 Features)

Recent work has considered the centrality of senders in their networks as an important signal of their local influence [71,72]. We constructed a social graph of the email interactions, with employees as the nodes, to implicitly capture the workplace relations and dynamics between employees. Then, we recomputed the social network metrics for the sender and receiver at the point when an email was sent based on their communication histories with each other and with everyone else.

If emails were exchanged between two employees, an edge was created between them, with its weight proportional to the number of emails. We calculated many centrality features that depicted the sender's role as an information broker. However, only PageRank and Betweenness centrality were found to be correlated with the probability of receiving an email reply, which are discussed below.

For $e_t : s \rightarrow r$, where $e_t$ = Email exchanged at time $t$, $s$ = Sender, $r$ = Recipient.

- **PageRank**: This measures the number of times the sender $s$ is encountered in a random walk over the social network:

$$C_p(x_r) = \alpha \sum A_{s,r} \frac{C_p(v_s)}{D_s^{out}} + \beta \tag{5}$$

where $x_r$ represents the recipient $r$. The adjacency matrix entry $A_{s,r}$ indicates the relationship from sender $s$ to recipient $r$. The term $C_p(v_s)$ represents the pagerank of the sender $s$. The out-degree $D_s^{out}$ represents the out-degree of the sender $s$.

- **Betweenness centrality**: This measures the number of the shortest paths connecting nodes that pass through a particular node. It identifies the degree to which senders act as conduits of information:

$$g(v) = \sum_{s \neq v \neq t} \frac{\sigma_{vt}(S)}{\sigma_{vt}} \tag{6}$$

where $\sigma_{tv}(S)$ is the number of shortest paths between $t$ and $v$ that pass through sender $s$, and $\sigma_{tv}$ is the total number of shortest paths from $t$ to $v$.

4.1.3. Personal Influence (10 Features)

Beyond the semantic characteristics of emails, the likeability of a sender among their supervisors and peers can reflect the individual differences that affect organizational responsiveness and communication quality. We included the average employees' **performance**

over three years before an email was sent, as this reflects their immediate supervisor's approval. The Luxury Standard dataset also includes the survey responses of a small subset of employees who rated their peers on a 0–5 scale on different dimensions, specifically **innovativeness**, **interactiveness**, **quality of inputs**, and **problem-solving aptitude**. They also rated their relationship with their peers for the **quality of friendship**, **communication**, and **collaboration frequency**, as well as the **level of avoidance**.

### 4.1.4. Linguistic Accommodation (3 Features)

The linguistic accommodation of the email was operationalized in terms of its similarity to the linguistic style of the sender and recipient. The equations below depict the procedure followed to generate these features. Prior work on examining online conversations typically included signals of linguistic accommodation as a signal of participants' interpersonal relationships [45,73,74]. First, to capture the linguistic styles of the sender and the recipient, the most recent four emails they sent to anyone else in the organization were concatenated. The 768-dimension BERT embedding of this concatenated text ($\phi_{BERT}$) represents the linguistic style of the sender $S$ or the recipient $R$ at time $= t$.

$$L_t^s = \phi_{BERT}\left(\left[e_{t-4}^s \mid e_{t-3}^s \mid ... \mid e_{t-1}^s\right]\right)$$
$$= \text{linguistic feature vector of } s \text{ at time } t \tag{7}$$

$$L_t^r = \phi_{BERT}\left(\left[e_{t-4}^r \mid e_{t-3}^r \mid ... \mid e_{t-1}^r\right]\right)$$
$$= \text{linguistic feature vector of } r \text{ at time} = t \tag{8}$$

Next, we computed the following to capture three measures of similarity between the current email, the sender, and the recipient:

$$s_c(e,s) = \frac{\phi_{BERT} \cdot L_t^s}{\|\phi_{BERT}\|\|L_t^s\|}$$
$$s_c(e,r) = \frac{\phi_{BERT} \cdot L_t^r}{\|\phi_{BERT}\|\|L_t^r\|} \tag{9}$$
$$s_c(r,s) = \frac{L_t^r \cdot L_t^s}{\|L_t^r\|\|L_t^s\|}$$

where $s_c(a,b)$ represents the cosine Similarity between vectors $a$ and $b$, and $s_c(e,s)$, $s_c(e,r)$, and $s_c(r,s)$ are the linguistic accommodation features we included in model training.

## 5. Results

### 5.1. Contribution Analysis

To address RQ1a and RQ1b, Figure 2 reports the model coefficients in generalized linear models trained to predict the likelihood of receiving a reply to a Sender's email. The data comprises the subset of the Luxury Standard dataset for which the personal influence features were available. Thus, the analyses are reported on 23 k rows instead of 600 k rows in the simple transformer implementation.

The plot comprises only the variables retained after feature selection because they had statistically significant correlations with the binary outcome (whether or not an email receives a reply). Overall, professional influence offers the biggest predictive power in positively and negatively affecting the likelihood of receiving a reply, where the sender's pagerank is a positive predictor ($\beta = 0.65$, $p < 0.001$) of receiving a reply. This implies that an increase in one standard deviation of a sender's pagerank is associated with a change of 0.65 in the log odds of the sender receiving a reply. In contrast, betweenness centrality is a negative predictor ($\beta = -0.67$, $p < 0.001$). This means that senders with a higher betweenness centrality (who often email individuals who are less connected with each other) reduce their log-odds' chances of receiving a reply by 0.67.

Similarly, among the stylistic features, we observe that emails using informal language ($\beta = 0.05$, $p < 0.05$) with cognitive processing ($\beta = 0.11$, $p < 0.001$) that mention words reflecting their psychological drives ($\beta = 0.06$, $p < 0.01$) are more likely to receive replies. In contrast, emails reflecting positive or negative emotions are less likely to do so ($\beta = -0.15$,

$p < 0.001$). After including the other variables in the model, the personal influence features were not statistically significant predictors of receiving email replies. Regarding linguistic accommodation, our findings confirm previous work regarding the importance of sender–recipient alignment for the likelihood of receiving a reply ($\beta = 0.17$, $p < 0.001$).

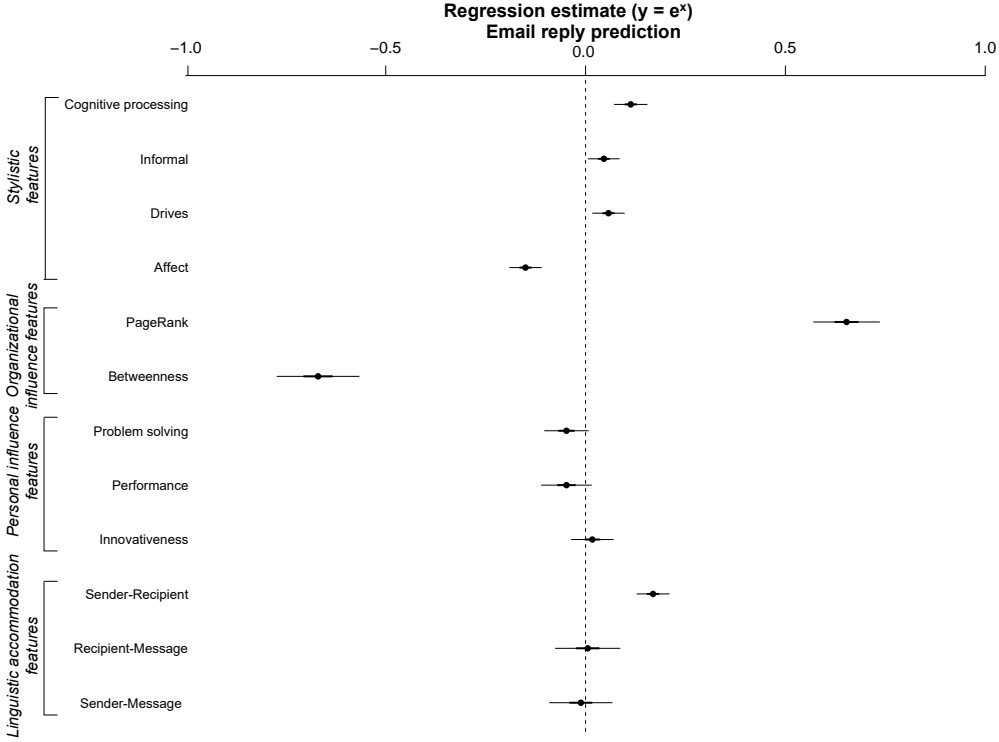

**Figure 2.** Effect sizes of the rescaled independent variables on whether an email receives a reply. The effects are reported as log-odds values ($y = e^x$).

### 5.2. Cross-Validation

To address RQ2, Table 2 reports the % accuracy, the macro F1-, and the minority F1-scores for the baseline models and EMMA. We see that the multimodal models perform better than all the baseline models, beating the most capable model (RoBERTa) on the macro F1-score by 13% and the minority F1-score by 25%. EMMA's performance on the minority class is by far the best of all the models evaluated, with a clear 19% lead on all the models, while it performs about as well as the other models on the majority class.

**Table 2.** Performance metrics for different models on the Luxury Standard dataset.

| Model | N = 665K | | | N = 23K | | |
|---|---|---|---|---|---|---|
| | **Accuracy** | **Macro F1** | **Minority F1** | **Accuracy** | **Macro F1** | **Minority F1** |
| | | | Baseline | | | |
| CNN | 0.75 | 0.54 | 0.24 | 0.73 | 0.54 | 0.24 |
| LSTM | 0.77 | 0.54 | 0.21 | 0.74 | 0.51 | 0.18 |
| BiLSTM | 0.75 | 0.55 | 0.25 | 0.74 | 0.54 | 0.23 |
| LSTM + Attention | 0.76 | 0.54 | 0.23 | 0.73 | 0.51 | 0.18 |
| BERT | 0.80 | 0.51 | 0.12 | 0.80 | 0.54 | 0.20 |
| RoBERTa | 0.81 | 0.54 | 0.19 | 0.81 | 0.54 | 0.18 |
| | | | EMMA: Email MultiModal Architecture | | | |
| EMMA with Network + Influence | | | | 0.82 | 0.65 | 0.40 |
| EMMA with Network + Influence + Context | | | | **0.84** | 0.66 | 0.41 |
| EMMA with Stylistic + Network + Influence + Context | | | | 0.83 | **0.67** | **0.43** |

### 5.3. Ablation Analysis

To address RQ3a, Table 3 reports the contribution of different feature combinations on the final predictive performance on the Luxury Standard dataset. Comparing the different multimodal models, we see that while features such as LIWC and Politeness, which capture different stylistic properties about the text, do increase the prediction of the minority class (minority F1-score = 0.50 vs. 0.18), they lead to an overall drop in accuracy (accuracy = 0.53 vs. 0.81). Adding influence features recovers performance accuracy (accuracy = 0.82 vs. 0.81), which is further improved by adding personal influence and linguistic accommodation features (accuracy = 0.84 in Column 3).

**Table 3.** Ablation study on feature contributions for the Luxury Standard dataset in the EMMA framework.

|  | RoBERTa | 1 | 2 | 3 | 4 |
|---|---|---|---|---|---|
| Stylistic features | - | ✓ |  |  | ✓ |
| Organizational influence features | - |  | ✓ | ✓ | ✓ |
| Personal influence features | - |  | ✓ | ✓ | ✓ |
| Linguistic accommodation features | - |  |  | ✓ | ✓ |
| Accuracy | 0.81 | 0.53 | 0.82 | 0.84 | 0.83 |
| Macro F1 | 0.54 | 0.70 | 0.65 | 0.66 | 0.67 |
| Minority F1 | 0.18 | 0.50 | 0.40 | 0.41 | 0.43 |

To address RQ3b and investigate the influence of dataset size on EMMA performance, we experimented with different stratified training samples from the Luxury Standard dataset. Figure 3 plots the accuracy, macro F1-score, and minority F1-score as a linear function of the dataset size. We observe that the blue line, depicting the EMMA variant with stylistic, social influence, and linguistic accommodation features, has the best and most consistent improvement in performance across all dataset sizes. The most considerable improvement in macro F1-score and minority F1-score is for EMMA models with network and social influence features.

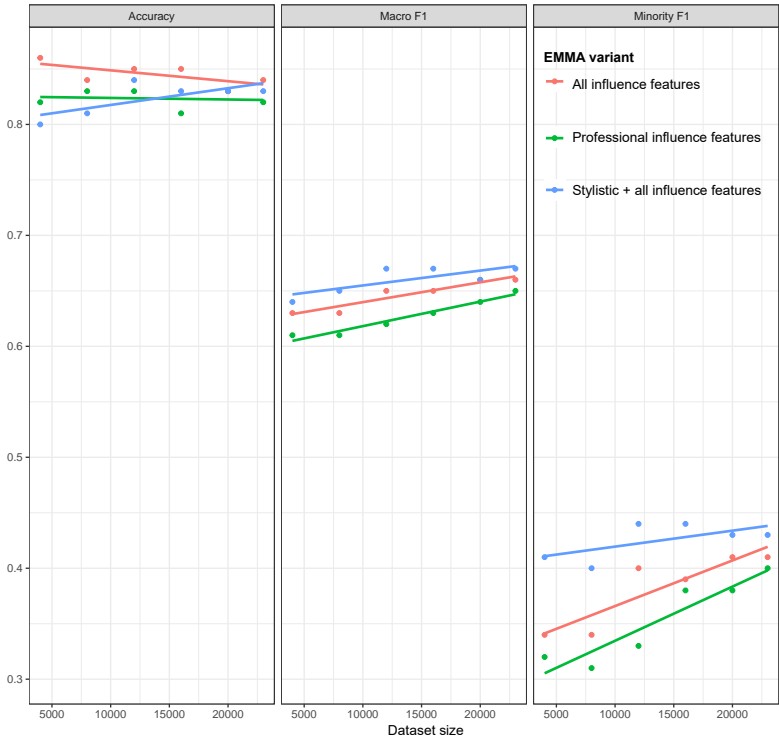

**Figure 3.** Data ablation analysis for EMMA variants. Across all dataset sizes, we see that a model with stylistic and influence features is consistent and superior to others.

### 5.4. External Validity

We examined whether EMMA is also effective at solving the same problem in other datasets and in other combinations that may have a subset of features. From Figure 4, we observe that the multimodal architecture models are notably better than the simple transformer models across all datasets, with an average improvement in accuracy of 12.5%. The maximum accuracy with EMMA models was seen in the Enron dataset, while the maximum improvement in minority F1-scores was seen in the Luxury standard dataset. The results inspire confidence in the EMMA architecture to model email data and networks for email reply prediction.

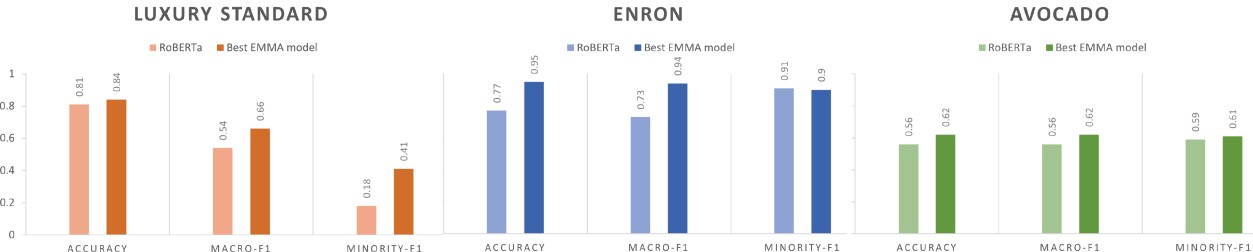

**Figure 4.** Generalizability analyses on the Enron and Avocado datasets. Among the three datasets, the biggest improvement in accuracy and macro F1-score is observed in the Enron dataset. The biggest improvement in minority F1-score is on the Luxury Standard dataset.

## 6. Discussion

Our analysis of email reply likelihood has unveiled several intriguing patterns that shed light on the dynamics of digital communication. One of the most salient findings pertains to the role of **professional influence** in determining the success of an email. Measures such as pagerank elucidate the connectivity and influence of nodes within their immediate and extended networks. The sender's pagerank emerged as a robust positive predictor of receiving email replies, suggesting that individuals with a higher pagerank, indicative of their prominence or influence in a network, are more likely to receive a response. This could also be attributed to the perceived authority or credibility associated with such individuals. While professional influence metrics shed light on the implicit connections between employees, personal influence metrics, derived from employee surveys, provide a snapshot of explicit relationships. This duality of explicit and implicit influences, especially when viewed through the lens of the weak tie theory, underscores the multifaceted nature of communication within organizational structures.

On the other hand, betweenness centrality, which represents senders who frequently email individuals less connected with each other, predicts a decreased likelihood of receiving a reply. These senders are, on average, likely to email many people outside their immediate clique and are accordingly less likely to receive replies. The finding offers a contrasting understanding of information brokers as necessary for diffusing information. Our findings suggest that brokers are unlikely important from the perspective of necessitating interpersonal communication, suggesting that much of their information sharing may constitute broadcasts rather than directed messages needing replies. While professional influence metrics shed light on the implicit connections between employees, personal influence metrics, derived from employee surveys, provide a snapshot of explicit relationships. This duality of explicit and implicit influences, especially when viewed through the lens of the weak tie theory, underscores the multifaceted nature of communication within organizational structures. Overall, our findings suggest that over and above the role of language and style, social network characteristics are critical to contextualizing community communication and anticipating future behavior.

Additionally, the stylistic features of the email content play a pivotal role. Informal language, cognitive processing, and words reflecting psychological drives enhance the probability of garnering a reply. However, emails laden with strong emotions deter pos-

itive or negative responses. This could be because emotionally charged emails might be perceived as less professional [2]. Some may not warrant a reply, e.g., an email that simply says "Thank you!". Others may warrant a meeting or a different form of closure, prompting recipients to delay or avoid responding.

The **cross-validation** results underscore the prowess of EMMA in predicting email replies. Our research indicates that even without explicit data, features can be extrapolated by transmuting the data into alternate formats, such as social graphs. Integrating these non-textual features with email content, mainly when processed through a multimodal transformer architecture, outperforms traditional text-only models regarding accuracy and F1 score. Notably, predictions concerning the minority class (emails unlikely to receive a response) witnessed a significant enhancement in EMMA's performance over the following best baseline (RoBERTa). It is noteworthy that EMMA not only outshines other models but does so with a significant margin.

Delving into the **ablation** analysis, it is evident that the interplay of various features can profoundly impact prediction accuracy. While specific stylistic properties enhance the prediction for the minority class, they can concurrently pull down the overall accuracy. However, introducing influence features acts as a counterbalance, restoring and augmenting accuracy. Overall, our findings suggest that over and above the role of language and style, social network characteristics are critical to contextualizing community communication and anticipating future behavior. In summary, the consistent performance of EMMA across different datasets and dataset sizes hints at a possible mechanism: a synergistic effect where the combination of diverse features provides a more comprehensive representation of the email dynamics, enabling more accurate predictions. This synergy might be the key to understanding and optimizing digital communication in the future.

## 7. Conclusions

We proposed the Email MultiModal Architecture (EMMA), which incorporates social factors into transformer finetuning for downstream task prediction. Our work offers a conceptual and empirical contribution toward understanding organizational communication behavior. First, at an epistemological level, our findings show that the social networks of organizations provide important signals for modeling workplace behavior. We reimagined a traditional 'multimodal' training setup for transformer finetuning by enriching our inputs with information about the social context of the email. Our findings suggest incorporating social network factors in the finetuning process yields better models and improves the accuracy and macro F1-scores compared to a purely text-based model in a RoBERTa-based training setup. EMMA offers an advantage over simpler deep learning and transformer baselines and generalizes well to two other email datasets. Training models on a multimodal feature representation also offers robust predictions for new contexts, such as the Enron and Avocado datasets.

Our work offers essential insights into the role and importance of individuals from a resource exchange perspective, confirming recent prior work on the trade-offs in bandwidth and information diversity from other scholars [5,15]. We found that employees with high organizational influence and linguistic accommodation are more likely to receive replies. On the other hand, employees who are, on average, likely to email many colleagues outside their immediate clique are less likely to receive replies. These insights can generalize to understanding communication behavior on other small-world networks, such as social media platforms, where research on the virality of content [75,76], hot streaks in user popularity [77], and their consequences for user and community behavior [78,79] all discuss the importance of social network features in understanding and modeling communication cascades.

Interactional metrics are instrumental in decoding the subtle relationships at play within a workplace, and data about friendships, collaboration frequency, and other interpersonal dynamics between the sender and receiver augment the context underlying an interpersonal communication thread. Employees can be more productive if they prioritize

their email behavior, which is also expected to improve interpersonal relationships and organizational efficiency. Much recent research has examined the psychological impact of email usage, with studies reporting that the volume of emails is a significant predictor of email stress [80]. Applying our findings can implement policies that allow individuals to check emails less frequently, thus leading to reduced stress [81], with related effects on self-esteem and locus of control [82] without potentially compromising efficiency. **Limitations and Future Work.** Representing each input as a concatenation of past communication, communication styles, and dynamically changing network influence is computationally intensive, but in future work, we will investigate approaches to recalibrate these scores iteratively rather than calculate them from scratch.

Many emails do not require a response or signal the end of the conversation, but they are treated alike in our model. It is helpful to include these email threads as they allow us to operate with the simplest assumptions and offer a model that can predict the outcome regardless of the email intent. It is promising that EMMA can predict their outcomes as well, as is observed with the considerable improvements in the minority F1-score as compared to the classical RoBERTa model.

Signals of personal influence were available for only a small sample of employees in the primary dataset. Future work calls for testing more signals of interpersonal and individual influence, which can be obtained from human resource records about the senders, in ways that offer more sophisticated representations of individual messages in context.

In future work, graph-based models, such as graph convolutional networks (GCNs) [83] and graph attention networks (GANs) [84], can be applied to model email datasets as a social network graph and formulate the problem as one of link prediction. We anticipate that the advantage of GNNs is that they can leverage information from the neighbor nodes in the social graph, similar to how a CNN exploits the spatial data in an image by looking at the adjacent pixels when sliding a window over the input. We also plan to explore other applications of the social network features to model the relationship dynamics evidenced in email interactions, as well as natural language generation applications for the linguistic accommodation paradigm we have proposed in the present work.

Furthermore, while our experiments demonstrate a paradigm that incorporates professional communication, it is easy to imagine how the paradigm might extend to other applications, where signals of authors' influence can enrich the unimodal understanding of textual data. For instance, future research could explore the application of our findings across various digital communication platforms, such as Slack, Microsoft Teams, or Google Meet, as well as other collaborative environments, such as GitHub and Google Documents. The idea also offers exciting opportunities for modeling and testing notions of information credibility and virality in a social media environment.

Regarding privacy considerations, it is important to acknowledge that using extensive personal and professional email data in our research raises significant privacy concerns. Ensuring such data's confidentiality and ethical use is paramount, and future studies should rigorously adhere to privacy guidelines and data protection regulations. For instance, technologies developed in [44] offer a way to fine-tune existing predictive models with an organization's data, which is increasingly being considered in recent paradigms that adapt Large Language Models for commercial or enterprise AI goals [85].

The practical application of the EMMA model in real-world settings may encounter technological and organizational hurdles. These include integrating the model into existing communication systems and the need for substantial computational resources. Future research should focus on developing more efficient and scalable versions of the model that can be easily adopted in various organizational contexts.

Additionally, our framework currently does not account for the role of emotional intelligence and non-verbal cues, which are crucial in human communication but are challenging to capture in a data-driven framework. Future iterations of the model should explore ways to incorporate these aspects to provide a more holistic understanding of communication dynamics.

**Author Contributions:** Conceptualization, K.J. and H.S.; methodology, K.J. and H.S.; software, K.J. and H.S.; validation, K.J. and H.S.; data curation, all authors; writing, K.J. and H.S. All authors have read and agreed to the published version of the manuscript.

**Funding:** This research received no external funding.

**Data Availability Statement:** The data for the primary analysis is unavailable due to privacy restrictions. The other datasets are in the public domain and are available through the Linguistic Data Consortium website. Please contact J.F. for further information about the primary dataset. The code used for training and testing the models is available at https://github.com/harshshah99/undergrad_theses/tree/main/EEE (accessed on 2 December 2023)

**Conflicts of Interest:** The authors declare no conflict of interest.

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
