# Peer review of "Building a Multimodal Classifier of Email Behavior: Towards a Social Network Understanding of Organizational Communication"

_information, doi:10.3390/info14120661_

Round 1

Reviewer 1 Report

Comments and Suggestions for Authors

The subject of this work is innovative and significant, but some aspects require a more precise presentation. Sections 1 and 2 could benefit from consolidation in specific areas, and redundancies, such as those in paragraph 4 of the introduction and section 2.2, should be eliminated.

A notable concern relates to the inadequate presentation of most of the equations; the causes of this problem should be remedied. Tables 2 and 3 are tiny in size, which prevents effective observation of the data. In addition, Table 2 is simply inserted without the corresponding references in the text.

A generalized problem is the lack of structural cohesion that is perceived throughout the article. To give an example, in section 4.1 reference is made to the RoBERTa model, but it is not until section 7.1 that a thorough explanation is provided. This pattern is repeated from sections 4 to 7, contributing to an overall sense of disorganization. Consequently, a thorough restructuring of the article is recommended, possibly involving the shortening of certain sections to eliminate unnecessary redundancies.

A notable deficiency is the absence of a quantitative analysis with other relevant work. The article does not compare its metrics with the results presented by some related works, which limits the contextual understanding of the contribution. Addressing this deficiency would improve the article and provide a more complete assessment of its conclusions.

Author Response

We thank the reviewer for their careful review and detailed comments. We have carefully applied their feedback to improve the paper in the following ways:

1. The Introduction section has been rewritten to remove redundancies

2. Equations have been edited

3. The sizes of Tables 2 and 3 have been increased and properly referenced in text.

4. The structure of the article has been improved to move the description of models and baselines together with the Experimental Setup section, followed by the Data and Feature Extraction.

5. With regards to a comparison with baselines, we have offered a comparison with standard methods that prior work has applied. We have clearly labeled these so that it is possible to interpret our benchmarks as a quantitative comparison with relevant work.

Additionally, we have also rewritten the Conclusion section to make it clearer and more impactful. Once again, we thank the reviewer for their time and efforts. Please see the attachment, which tracks the changes in the revision from the original submission.

Reviewer 2 Report

Comments and Suggestions for Authors

The article is particularly pertinent in the era of remote work, addressing the crucial role of email in professional communication. By introducing the innovative Email MultiModal Architecture (EMMA), it takes a novel, data-informed approach to predict email behaviors, incorporating social networks and interpersonal relationships. The research's strength lies in its large-scale analysis of a comprehensive dataset of 0.6 million emails from thousands of employees, which substantiates its findings. Additionally, the practical applications of the study promise to enhance email communication efficiency within organizations, and the model's demonstrated accuracy across various datasets suggests that these improvements could be widely applicable, offering significant benefits to businesses aiming to optimize their communication strategies.

On the other hand, the article's focus on the technical development of a multimodal classifier could pose comprehension challenges for those unfamiliar with computational models. There are also privacy considerations to be taken into account, given the use of extensive personal and professional email data in the research. Practical application of the EMMA model in real-world settings may encounter technological and organizational hurdles, potentially limiting its immediate adoption. The article's emphasis on email might also be seen as slightly out-of-step with the growing diversity of communication tools used in modern workplaces. Lastly, the reliance on data for communication prediction risks downplaying the inherently human elements of communication, such as emotional intelligence and non-verbal cues, which are crucial yet difficult to capture in a data-driven framework.

To mitigate the potential drawbacks, the article could broaden its narrative by discussing the integration of the EMMA model with other forms of digital communication platforms, thereby acknowledging the multifaceted nature of modern organizational communication. Privacy concerns can be addressed by outlining the ethical considerations and data anonymization processes used in the study, reassuring readers about the commitment to data protection. Simplifying the explanation of the multimodal classifier and its workings could make the article more accessible to a non-technical audience. Further, highlighting case studies or scenarios where the human aspect of communication complements the predictive model would underscore the value of human judgment in conjunction with data-driven insights. Lastly, practical implementation strategies could be outlined, perhaps suggesting phased adoption or the development of user-friendly interfaces to facilitate the uptake of the EMMA model in real-world settings, making the transition smoother and more acceptable to organizational stakeholder.

Author Response

We thank the reviewer for their careful feedback. We have addressed their concerns in rewriting major parts of our manuscript, such as:

  1. In the Methods section, rewriting the description of the experiments and reordering the sections to make the article easier to read and follow
  2. In the Conclusion section, offering ways to integrate the framework with the growing diversity of communication tools used in modern workplaces.
  3. In the Limitations section, discussing privacy considerations of the use of extensive personal and professional email data in the research.
  4. In the Limitations section, discussing the practical application of the EMMA model in real-world settings may encounter technological and organizational hurdles
  5. In the Limitations section, discussing the role of emotional intelligence and non-verbal cues, which are difficult to capture in a data-driven framework.
  6. In the Limitations section, offering suggestions for how to incorporate the framework in practical contexts.

We thank the reviewer again for their time. Please see the attachment, which tracks the changes in the revision from the original submission.

Reviewer 3 Report

Comments and Suggestions for Authors

In this paper, the authors propose EMMA, a Multimodal Classifier of EMail behavior. EMMA uses data related to the email's sender's social network, performance metrics and peer endorsement to predict the probability of receiving an email response.

The topic considered by the authors is quite interesting; however, they should better motivate at the beginning of the introduction why this topic is relevant and deserves further investigation.

The weakest part of the paper is related work; this should be greatly improved. Indeed, despite the huge, indeed excessive, number of citations in the paper (which I suggest should be reduced) only a few papers are considered in related work. There are many other approaches both related to email behavior and concerning the role of social networks that the authors should consider. Regarding the first topic, authors should consider papers related to user sentiments (e.g., the paper "A framework for investigating the dynamics of user and community sentiments in a social platform"). On the other hand, regarding the second topic, authors should consider papers related to information dissemination (e.g., the paper "An approach to detect backbones of information diffusers among different communities of a social platform").

Comments on the Quality of English Language

The English of the paper is good

Author Response

We thank the reviewer for the insightful comments.

In the revised version we have:

  1. Rewritten parts of the introduction to better motivate the importance of the problem
  2. Reviewed the papers we have cited and removed any which did not tie in with our focus on organizational behavior, email communication behavior, or with the machine learning frameworks we have applied
  3. Added references to a few more papers that are more relevant to our problem, including those mentioned by the reviewer.

We have also rewritten our Method and Conclusion sections to address the reviewer's scores. Once again, we thank the reviewer for their time and feedback. Please see the attachment, which tracks the changes in the revision from the original submission.

Round 2

Reviewer 1 Report

Comments and Suggestions for Authors

The authors have addressed all my comments. I think this paper can be accepted for publication in its present form.

Comments on the Quality of English Language

Overall, this paper is well-written.

Reviewer 3 Report

Comments and Suggestions for Authors

The authors have striven to comply with my suggestions. Therefore, in my opinion, the paper can be accepted.

Comments on the Quality of English Language

The English of the paper is good